# Genetic Variation in *Caenorhabditis elegans* Responses to Pathogenic Microbiota

**DOI:** 10.3390/microorganisms8040618

**Published:** 2020-04-24

**Authors:** Yuqing Huang, Jan E. Kammenga

**Affiliations:** Laboratory of Nematology, Wageningen University, Droevendaalsesteeg 1, 6708 PB Wageningen, The Netherlands

**Keywords:** *Caenorhabditis elegans*, microbiota, genetic variation

## Abstract

The bacterivorous nematode *Caenorhabditis elegans* is an important model species for understanding genetic variation of complex traits. So far, most studies involve axenic laboratory settings using *Escherichia coli* as the sole bacterial species. Over the past decade, however, investigations into the genetic variation of responses to pathogenic microbiota have increasingly received attention. Quantitative genetic analyses have revealed detailed insight into loci, genetic variants, and pathways in *C. elegans* underlying interactions with bacteria, microsporidia, and viruses. As various quantitative genetic platforms and resources like *C. elegans* Natural Diversity Resource (CeNDR) and Worm Quantitative Trait Loci (WormQTL) have been developed, we anticipate that expanding *C. elegans* research along the lines of genetic variation will be a treasure trove for opening up new insights into genetic pathways and gene functionality of microbiota interactions.

## 1. *Caenorhabditis elegans* Responses to Microbiota

*Caenorhabditis elegans* (Nematoda; *Rhabditidae*) is a cosmopolitan hermaphroditic nematode that thrives on bacteria in rotting plant material [1,2]. It was introduced as a tractable species for biological research during the early 1970s [3], and since then, it has been established as a widely used model species in biological research. As a bacterivorous nematode, *C. elegans* populations can easily be reared in the lab. Research using *C. elegans* has mainly investigated nematodes kept under axenic laboratory conditions (i.e., populations are maintained in petri dishes at 20 °C on agar plates seeded with *Escherichia coli* as a food source) that are devoid of interactions with natural pathogens. Microbiota provide *C. elegans* with nutrients, but they may also be pathogenic, leading to decreased growth and fitness of nematodes [4]. Over the past few years, there has been a growing interest in unraveling the interaction between *C. elegans* and its pathogenic microbiota, like microsporidia, bacteria, and viruses, with the aim to better understand *C. elegans* ecology away from *E. coli* [5,6]. The *C. elegans* microbiota community normally inhabits or is closely associated with the nematode [7]. Recently, it was reported that the microbiota are able to synthesize essential nutrients for *C. elegans* and that the microbiota assembly is highly dynamic in time and species composition [8]. Understanding the interactions between *C. elegans* and microbiota may help further in gaining a deeper insight of microbiota interactions in other species. This comes at an important moment since it is increasingly realized that microbiota play in an important role in metazoan health [9].

## 2. Genetic Variation in *Caenorhabditis elegans* as the Basis to Understand Microbiota Interactions

The vast majority of *C. elegans* microbiota studies are host-pathogen interaction studies and have been limited to elucidating the molecular mechanism of host-pathogen interaction using a single *C. elegans* wild type strain, the canonical strain Bristol N2 [10]. Yet, studying genetically variable *C. elegans* strains will provide a broader view on the genetic architecture of host-pathogen interactions (Table 1). Over the past two decades, *C. elegans* has been established as a platform for studying the genetic architecture of complex traits using quantitative genetic analyses based on genetic variation [11,12]. A range of complex phenotypes have been mapped to loci and genetic variants using genetic linkage studies or genome-wide association approaches [13]. The *C. elegans* Natural Diversity Resource (CeNDR) was established to collect all genetic and genomic studies based on wild strains, thus facilitating genome-wide association studies (GWAS) [14]. It is a comprehensive data platform showing the genetic variation of *C. elegans* worldwide and facilitating genome-wide association analyses. Worm Quantitative Trait Loci (WormQTL) was recently launched, a platform for systems genetics in *C. elegans* [15]. Next to this, genetic resources such as recombinant inbred line (RIL) populations and introgression line (IL) populations derived from wild type Bristol N2 and Hawaii CB4856, or other wild strains, were constructed to unravel quantitative trait variants or genetic architectures [16]. For instance, two multiparent RIL panels were constructed derived from either four or eight parental wild strains. The quantitative trait loci (QTL) mapping panel *C. elegans* eight parental experimental evolution (CeMEE) contains more than 500 RIL. CeMEE has already been applied in *C. elegans* research to understand the genetic basis of complex phenotypes, like fertility and the hermaphrodite body size during reproduction [11,13,17]. Another four parental population of recombinant inbred lines was derived from wild type strains that together captured genetic variation on a local scale [13]. Several studies using a range of these genetic resources have been used to study the genetic variation regarding the *C. elegans*–microbiota interactions (Table 1). These studies will be discussed in more detail below.

## 3. *Caenorhabditis elegans*–Bacteria Interactions

Of all *C. elegans* microbiota groups, bacteria are the most well studied. Bacteria may serve as food and/or may be pathogenic to *C. elegans*. The most abundant bacteria studied include *Gammaproteobacteria* (*Enterobacteriaceae*, *Pseudomonaceae*, and *Xanthomonodaceae*) and Bacteroidetes (*Sphingobacteriaceae*, *Weeksellaaceae*, and *Flavobacteriaceae*) complemented with a few *Acetobacteriaceae* [25,26]. *Proteobacteria*, *Bacteroidetes*, *Firmicutes*, and *Actinobacteria* are bacterial phyla found living in the same habitat as *C. elegans* [27]. *C. elegans* populations thrive well on these bacteria, and some of these bacteria may even increase resistance to abiotic and biotic stress factors, including pathogen infection [25,28]. *C. elegans* is able to avoid the pathogenic bacteria, *Pseudomonas aeruginosa*, through aversive olfactory learning. This ability could be transmitted to next generations under the function of P-element Induced WImpy testis (Piwi) Argonaute homolog Piwi Related Gene (PRG-1), thus improving the fitness of the progeny [29]. Besides, innate pathways are also under investigation [30]. *Pseudomonas vranovensis* is a natural pathogen of *C. elegans*, parental exposure of *C. elegans* to *P. vranovensis* upregulates the cysteine synthases CYSL-1 and CYSL-2 and the regulator of hypoxia inducible factor RHY-1, which promotes the resistance of progeny to infection [31]. Besides variant strains, sex differences in *C. elegans* also have impact on the resistance to *Bacillus thuringiensis* [32]. Under infection pressure from *B. thuringiensis*, males of *C. elegans* have lower resistance and survival than hermaphrodites. Next to this, proteome levels in *C. elegans* differ depending on the type of bacteria that they feed on [10], like *Ochrobactrum* spp. (MYb71, MYb237) or *E. coli* OP50.

Schulenburg and Muller (2004) showed that genetic variation affects infection of *C. elegans* by a parasitic strain of Gram-positive *B. thuringiensis* using ten wild strains [18]. Their analyses revealed genetic variation for behavioral responses as an important defensive factor in *C. elegans*. Here, the behavioral trait comprised evasion and reduced parasite ingestion. As for the gene level, different quantitative genetic studies have identified the neuropeptide Y receptor gene, *npr-1*, as a major pleiotropic determinant regarding bacterial food intake. Natural variation in *npr-1* was first reported to modify social behavior and responses to food [33]. Later, it was reported that polymorphic *npr-1* affected behavioral responses to the Gram-negative pathogen *Pseudomonas aeruginosa* by means of oxygen-dependent behavioral avoidance rather than direct regulation of innate immunity [19]. Nakad et al. (2016) investigated the role of *npr-1* in *C. elegans* exposed to *B. thuringiensis* and *P. aeruginosa* using QTL analyses of behavioral immune defense and RNAseq-based transcriptomics using recombinant inbred lines and introgression lines [34]. Several QTL were detected, including one on chromosome X harboring *npr-1*. The wild type N2 allele was found to be associated with reduced defense against *B. thuringiensis* and had an opposite phenotype to that previously mapped for the N2 *npr-1* allele against *P. aeruginosa*. Global gene expression profiling suggested that *npr-1* plays an important role against both pathogenic bacteria through p38 Mitogen-Activated Protein kinases (MAPK) signaling, insulin-like signaling, and C-type lectins.

The reduced sensitivity to *P. aeruginosa* was associated with the induction of oxidative stress genes and activation of GATA transcription factors. In contrast, oxidative stress gene suppression combined with the activation of Ebox transcription factors determined *B. thuringiensis* effects. Like *npr-1*, genetic variation in other genes related to neuronal functions was found to play a key role in detecting food related cues, like tyramine receptor gene *tyra-1*, encoding a G protein-coupled catecholamine receptor [35]. Interestingly, *npr-1* was found to control a range of phenotypic differences through behavioral avoidance of ambient oxygen concentrations to growth and physiology. Together, these results show that genetic variation in *npr-1* gives rise to pleiotropic effects that are associated with bacterial food [36].

Next to *npr-1*, Chang et al. (2011) found polymorphisms in a Homologous to the E6–AP Carboxyl Terminus (HECT) domain-containing E3 ubiquitin ligase, HECW-1. Two polymorphisms close to residues of HECW-1 each affected *C. elegans* behavioral avoidance exposed to *P. aeruginosa* [37]. The resistance of *C. elegans* to *P. aeruginosa* is achieved by avoidance instead of physiological stress pathways. Neuronal analysis pointed out that HECW-1 functions in two of sensory neurons against *P. aeruginosa* through inhibition of the neuropeptide receptor NPR-110.

In *C. elegans*–bacteria interaction studies, QTL analysis was not only used to study *npr-1*, but also to detect other genetic variants. For example, it was found that the ability of *C. elegans* to sense bacteria also differed among the different strains of *C. elegans*. The odor of *Serratia marcescens* is more attractive to Bristol N2 than Hawaii CB4856. In this case, QTL analysis helped to define the region that interacts and affects *S. marcescens* preference based on RILs [20]. In summary, genetic variation study of *C. elegans*–microbiota systems were mostly researched in bacteria until now. The results varied from gene level to behavior level, expanding the insight into different host mechanisms in *C. elegans* against bacterial pathogen.

## 4. *Caenorhabditis elegans*-Microsporidia Interactions

Microsporidia are a group of eukaryotic intracellular pathogens comprising more than a thousand species that are associated with a wide host range [38]. *Nematocida parisii* was the first microsporidian species found to infect *C. elegans* and causes lethal infections. The mechanism of microsporidia to invade host tissue is mainly achieved by restructuring multicellular host tissues into syncytia and proliferating cells across the host tissue cells before forming spores [39]. As *N. parisii* has been isolated from wild *C. elegans*, *C. elegans*–*N. parisii* interactions have become an important model for studying both the infection process of pathogen and response of the host. The response of *C. elegans* to *N. parisii* involves a transcriptional response named intracellular pathogen response (IPR) that can be triggered by proteotoxic stress to improve the resistance to the stress [40]. This response results in upregulation of 80 so-called IPR genes regulated by genes *pals-22* and *pals-25* [41]. Whilst studying natural pathogens of *C. elegans* including microsporidia, it was found that IPR involves gene *pals-22* and *pals-25* which play an essential role in combatting pathogens. Mutation of *pals-22* showed increased resistance to microsporidia infection as well as virus infection. However, mutants of *pals-25* showed suppressed phenotypes caused by mutation in *pals-22*, including increased IPR gene expression, thermotolerance, and immunity against natural pathogens. Gene *pals-22* was proved to repress expression of some ubiquitin ligase complex components and other IPR genes [41].

Microsporidia are transmitted horizontally and replicate predominantly in intestinal cells in *C. elegans*. However, among different species of microsporidia, host range [42], virulence [39], and tissue tropism strongly differ [43]. For example, *N. parisii* only invades and replicates in the intestinal cells of *C. elegans*, while *Nematocida displodere* is able to infect multiple host tissues [43]. Besides, some IPR genes regulated by *N. parisii* in *C. elegans* were hardly induced by *Nematocida ausubeli*, suggesting it may evolve to counteract with the host response [42]. In the meantime, among different species of nematodes, their responses to microsporidia are not the same as well. For instance, after three days of infection, for *C. elegans* and *C. briggsae*, 50% of the worms were infected, while for *Oscheius tipulae*, no infectious behavior was found [42].

Genetic variation in epithelial immunity was found in *C. elegans* after infection of the intestines by *Nematocida ironsii* [21,22,23]. Wild type CB4856 displayed a higher clearing and resistance to infection than wild type Bristol N2, and this difference was specifically found in their young larvae. It was concluded that this antagonistic pleiotropic effect in CB4856 enables a higher progeny later in life and leads to a selective advantage under laboratory conditions. In the study, resistance was mapped to four genomic loci, and two of these loci influencing resistance were confirmed [23]. Overall, the quantitative genetic study revealed that resistance to microsporidia early in the life of *C. elegans* is a complex trait associated with higher reproductive output later in life, thus enhancing overall fitness. Currently, only a few genetic variation studies of microsporidia infections in *C. elegans* have been conducted. More research into the genetic variants affecting epithelial tolerance is required since the epithelium plays a major role in host–pathogen interactions.

## 5. *Caenorhabditis elegans*–Virus Interactions

Studies on *C. elegans*–virus interactions were hampered by the lack of a naturally infectious virus until 2011 when Orsay virus was found [24,44]. Orsay virus is a single-stranded RNA (ssRNA) positive-strand virus related to the family Nodaviridae [45]. It is the first virus found to naturally intracellularly infect and complete its whole life cycle in *C. elegans* intestinal cells [24,46]. Orsay virus consists of two segments: the RNA1 segment encodes viral RNA–dependent RNA polymerase (RdRP), and the RNA2 segment encodes capsid protein and one additional open reading frame (ORF) of 332 amino acids at 3′ end called delta (ORF δ) [46,47]. ORF δ is important for nonlytic viral egress [48,49]. Contrary to most Nodaviruses, Orsay virus does not contain a RNA-3 translated suppressor for RNAi, and RNA2-encoded protein also lacks RNAi suppression function [50].

Many pathogens of *C. elegans* share the same target tissue, the intestine [47]. Orsay virus is no exception. The intestine in *C. elegans* consists of 20 cells, holding a proportion of one third of somatic mass and is essential for absorbing nutrient and producing food derived macromolecule [46]. It was found that intestinal cells showed high susceptibility when infected by virus, while few cells from the entire host tissue were infected [46]. Orsay virus affects the intestinal region of nematode which is mainly reflected by the disappearance of cell structures like storage granules and extensive convolutions of the apical intestinal border. However, due to lack of the adaptive immune system and specialized immune cells, the defense against virus infection critically depends on epithelial cells. Interestingly, the key anatomical features of the epithelium are conserved between *C. elegans* and mammals [51], which allows studies in *C. elegans* to be translated to higher species.

Orsay virus infection also triggers the IPR pathway. The exposure to Orsay virus led to upregulation of the Skp, Cullin, F-box (SCF) ubiquitin ligase components, such as the cullin ortholog CUL-6. The ubiquitylation components, the proteasome, and the autophagy pathway are all important for defense against viral infection [52]. However, although *C. elegans*–Orsay virus and *C. elegans*–microsporidia interaction pathways both belong to IPR, there is a difference between the two pathogens. The *C. elegans* retinoic acid-inducible gene (RIG)-I-like receptor DRH-1 was proved to be necessary for inducing IPR when infected by Orsay virus, but not for microsporidia infection or proteotoxic stress [40]. Differentially expressed genes identified responsible for immune response within *C. elegans*–microsporidia and *C. elegans*–Orsay virus interactions were also not the same.

The intraspecies susceptibility to Orsay virus of *C. elegans* is highly variant [53,54], as different *C. elegans* strains have distinct susceptibilities when faced with Orsay virus infection. Studies into the genetic variation to Orsay virus infection revealed that DRH-1, encoding a RIG-I-like helicase, is required for the initiation of an antiviral RNAi pathway and the generation of virus-derived siRNAs (viRNAs) [53]. Wild type Bristol N2 was more resistant than strain JU1580 as it accumulates 50–100 fold less viral RNA inside at young adult stage [24]. The difference was also shown for L2–L4 nematodes, as the final viral load of JU1580 was significantly higher than Bristol N2. It was noted that there were offspring right before measuring viral load. Thus, maybe N2 progeny might lose vertical transmitted infection after several generations, while JU1580 progeny remains infected [54]. In any case, the difference regarding viral susceptibility is affected by genetic variation, thus research on polymorphic loci is essential for identification of causal loci. GWAS analysis is often applied for the detection of polymorphic variants which could link genetic variation with phenotypes. Subsequently, inbred lines and introgression lines can be generated by crossing different wild types of *C. elegans* [55]. For instance, experiments using GWAS and introgression lines proved that a *drh-1* polymorphism is involved in viral susceptibility [53]. In summary, based on *C. elegans* genetic variation studies involving Orsay virus, insight into antiviral immunity in *C. elegans* was developed and more detail in viral infection and host response was revealed.

## 6. Conclusion and Future Outlook

Quantitative genetic analyses into different *C. elegans* genotypes of pathogenic microbiota interactions can be complementary to studying interactions in a single genotype. Studying microbiota interactions across multiple genotypes allows for unraveling the genetic architecture of these interactions.

Mapping of genetic variation is an essential tool in investigating the genetic basis of complex traits that underlie the interaction between *C. elegans* and its associated microbiota. Resources like CeNDR, in combination with QTL studies, could provide more insight of gene function and evolutionary studies. To overcome the limitations of RILs that were derived from a combination of two parental strains (usually Bristol N2 and Hawaii CB4856), multiparent populations like CeMEE and the four parental population panel can improve the power of detecting natural genetic variants [13].

In the future, with the application of advanced methods of quantitative genetics study, genetic variation of *C. elegans* could be explored more to reveal the genetic architecture of variants relating to complex microbiota related traits. This may promote the study of the underlying molecular genetic mechanisms at, for instance, the metabolic and protein levels. Further, it may aid to understand how microbiota influence the fitness of *C. elegans* and how functional genes affect the microbiota interactions. For example, a recent study on nematode trapping fungi (NTF) demonstrated the trap induction, involving G protein β–subunit Gpb 1, plays as fitness character in NTF family [56]. A next step would be to combine different microbiota, like bacteria and viruses, in order to assess their impact simultaneously. Together, they could be exploited to obtain knowledge on both molecular and evolutionary aspects of host-microbiota interactions, including but not limited to microsporidia, bacteria, and viruses. Thus, it would provide more information on how organisms in nature adjust and evolve in the interaction with its microbiota.

## Figures and Tables

**Table 1 microorganisms-08-00618-t001:** Quantitative genetic studies of *C. elegans*–microbiota interactions.

Microbiota	Species	Phenotypes	Strains of *C. elegans*	Reference
Bacteria	*B. thuringiensis*	Behavior response (evasion and reduced parasite ingestion)	Ten wild strains; RILs and ILs	[18]
	*P. aeruginosa*	Behavior response (oxygen-dependent behavioral avoidance)	RILs and ILs	[19]
	*S. marcescens*	Odor attractiveness	CB4856 and N2	[20]
Microsporidia	*N. ironsii*	Ability of clearing infection; initial colonization of *Nematocida*	CB4856 and N2	[21,22]
	*N. ironsii*	Resistance in young L1 larvae	CB4856 and N2	[21,22,23]
Virus	Orsay virus	Susceptibility	N2 and JU1580; ILs (GWAS)	[24]

RIL: recombinant inbred lines; IL: introgression lines; GWAS: genome wide association study.

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
