# Peer review of "Genetic Variation in Caenorhabditis elegans Responses to Pathogenic Microbiota"

_microorganisms, 2020, doi:10.3390/microorganisms8040618_

Round 1

Reviewer 1 Report

Huang and Kammenga have reviewed natural genetic variation in C. elegans response to microbiota. It is very comprehensive, however, so many abbreviations which are suggested to be listed by full-names and explained for those readers are not familiar with special terms.

It is also suggested to put few sentences to described the nematode-trapping fungi. Or at least add a reference of Yang et al., (2020, PNAS March 24, 117:6762-6770) in line 224.

Minor

Line 167, explain what is “one additional ORF”. At least let readers know how large the protein (number of amino acid residues) is and possible function.

Line 291: Delete (Trends in ……April 2, 2016) which is redundant.

References should be in a consistent form. References 1, 3, and 5 (and so on) is one form while references 2, 4, 6, 7….in aother form.

Reviewer 2 Report

Review of “Natural genetic variation in Caenorhabditis elegans responses to microbiota” by Huang and Kammenga
Overview
In this review, Huang and Kammenga summarize recent findings on C. elegans interactions with different types of microorganisms. Specifically, they review genetic variations of C. elegans that were found to play a role in the response to pathogenic microorganisms.
With some important modifications (see below), this manuscript could be a valuable short summary of our current knowledge about host-pathogen interactions using C. elegans as a host model organism.
Major criticisms
1. The title of this review is misleading for two reasons. First, the main focus of this review are host-pathogen interactions. This is understandable because there is very little data about non-pathogenic species of the C. elegans microbiota. However, it should be clear in the title that this review is almost completely about analyzing genetic variations in the response of C. elegans to different pathogens, not symbionts. Secondly, not all the discussed genetic variations are natural, as stated in the title. For example, the authors summarized findings about IPR regulators, which were not identified as natural genetic variations.
2. The organization of this review could be improved. It is unclear why the authors chose the current order of subjects in the manuscript (interactions with microsporidia, bacteria, virus), especially because there is an overlap in the C. elegans response to both microsporidia and viruses.
3. The strain of N. parisii that was found to be cleared in CB4856 but not in N2 animals was recently renamed into N. ironsii, because whole genome sequencing studies showed that this is a different species from the originally isolated N. parisii strain (Reinke 2017, Balla 2019).
Minor criticisms
1. Claiming that microsporidia destroy tissues before they produce spores is not completely correct, as only localized destruction of the membranes between two neighboring cells was reported. Complete tissue destruction would lead to death, because intestinal cells of C. elegans cannot regenerate.
2. Row 72: “interactions has” should be “interactions have”
3. Row 75: “improve the ability against the stress” – this is an incomplete statement that does not make sense as written.
4. Authors are using medical terms in describing phenotypes, which is not standard for C. elegans field. Please avoid using terms “symptoms” and “individuals”.
5. Claiming that IPR genes are not induced by N. ausubeli should be stated in a more cautious way, because not all IPR genes were analyzed in the referenced study, and a low induction was observed for some of the analyzed genes.
6. Row 142: “other neuro related genes” should be rephrased.
7. Row 167: the authors should state that the other ORF of RNA2 encodes a delta protein, which is important for viral egress.
8. Row 172: “absorbing nutrition and producing macro molecule” – this needs to be modified. Nutrients, not nutrition are absorbed. What is the macro molecule that authors are referring to?
9. Row 176: “Proper immune system” – C. elegans has many conserved components of the innate immune system, so the authors need to be more precise about what they consider to be a proper immune system. Adaptive immunity? Professional immune cells?
10. In several instances phrase “susceptibility against” was used. These two words cannot go together – animals can only be susceptible to pathogens.
